

# Can high-resolution convection-permitting climate models improve flood simulation in southern Quebec watersheds?

Behmard Sabzipour[1], Philippe Lucas-Picher[1], Richard Turcotte[2], Gabriel Rondeau-Genesse[3]

[1]Centre pour l'Étude et la simulation du climat à l'échelle régionale (ESCER), Département des sciences de la Terre et de l'atmosphère, Université du Québec à Montréal, Montréal, H3C 3P8, Canada
[2]Direction principale de l'expertise hydrique (DPEH), Ministère de l'Environnement et de la Lutte contre les changements climatiques, de la Faune et des Parcs (MELCCFP), Québec, G1R 5V7, Canada
[3]Ouranos, Montréal, H3A 1B9, Canada

*Correspondence to*: Behmard Sabzipour (sabzipour.behmard@courrier.uqam.ca)

**Abstract.** In August 2024, Montreal and its surroundings, located in the south of the Quebec province, experienced one of its most destructive meteorological events in history, associated to the remnants of the tropical storm Debby, according to the Insurance Bureau of Canada (Published on 2024, September 13). With climate change, the frequency and intensity of extreme weather events are expected to increase, explaining why government and private sectors, particularly insurance companies, requires enhancing their preparedness.

Recent studies highlighted the potential of high-resolution climate models (with grid sizes smaller than 4 km) to improve precipitation extremes at sub-daily timescales. This study focuses on heavy rainfall events during the warm season, comparing outputs from the latest Canadian Regional Climate Model (CRCM6/GEM5) at 12 km and 2.5 km resolutions. For the first time, we estimated that the CRCM6/GEM5-2.5km better captured the intensity of extreme hourly rainfall events compared to the CRCM6/GEM5-12km, aligning more closely with weather station data.

To assess whether this added value extends to hydrological modeling, we used a lumped hydrological model to simulated water flows at an hourly time step for 11 basins located over southern Quebec for the period 2001–2018. For most basins, summer-fall peak flows simulated using the CRCM6/GEM5-2.5km had lower biases compared to those simulated with the CRCM6/GEM5-12km. These findings emphasize the importance of high-resolution climate models in improving extreme event simulations, which is essential for better risk assessment and adaptation strategies in a warming climate.

## 1 Introduction

In August 2024, southern Quebec, Canada—specifically Montreal and its surrounding areas, home to over 4 million residents—experienced an exceptional precipitation event. Within a span of less than 24 hours, many weather stations over Montréal recorded >150 mm of rainfall on August 9 2024, far surpassing the region's monthly average of approximately 90 mm (based on the 1991–2020 mean) and the previous daily record of around 100 mm. This extreme rainfall event led to a widespread flooding, which is the most frequent and destructive natural disaster in Canada (Buttle et al., 2016). Flooding in



Canada is not only a recurring threat, but also a significant economic burden, with an average annual cost of $1.4 billion in flood-related damages (Morin et al., 2025).

In response to the increasing frequency of such extreme events, the Quebec government launched INFO-Crue, a comprehensive initiative to study and mitigate the impacts of flooding throughout the province (Lavoie & Turcotte, 2020).

This initiative underscores the growing need for more comprehensive flood risk assessment and management strategies. With the increasing frequency and intensity of extreme weather events, driven mainly by global warming (Guerreiro et al., 2018; Myhre et al., 2019; Martel et al., 2021; Moustakis et al., 2021), flood risks are increasing. Such events, not only disrupt communities, but also pose a growing threat to human lives and infrastructure, with flooding becoming more widespread and severe globally (Buttle et al., 2016; Wang et al., 2022).

General Circulation Models (GCMs), also known as Global Climate Models, have been instrumental in studying atmospheric phenomena and climate change for over 60 years (Smagorinsky et al., 1965; Manabe et al., 1965). These models are designed to simulate and understand the Earth's climate system, including the complex interactions between the atmosphere, oceans, land surface, and ice (Covey et al., 2003). Over the past six decades, GCMs have continuously improved, particularly in their ability to represent the Earth's climate system in greater detail. Early progress in providing 45 finer resolutions climate simulations emerged from studies such as Dickinson et al. (1989) and Giorgi (1990), who introduced limited-area models, or nested regional climate models (RCM).

Driven by the need to refine climate simulations and improve their realism, the development of RCMs began nearly three decades ago (Giorgi and Mearns, 1999). RCMs, using limited-area domains that are driven by GCM simulations at their lateral boundaries, can nowadays produce multi-decadal climate simulation with grid spacings of around 10 km 50 (Rummukainen, 2010; Giorgi, 2019). This increased resolution allows RCMs to better represent topography and atmospheric processes occurring at the 10-100 km spatial scale, such as intense precipitation events associated to complex terrain and coastal processes, as seen in regions such as the Mediterranean basin (Ruti et al., 2016). The higher resolution of RCMs has proven particularly effective in improving the spatial patterns of precipitation (Rauscher et al., 2010; Maraun and Widmann, 2015), which is crucial for accurately simulating extreme weather events and hydrological impacts.

Regional Climate Models (RCMs) are particularly valuable for flood studies due to their higher resolution, which enables them to better capture extreme precipitation events (Prein et al., 2015; Casanueva et al., 2016). These models are especially important for flood simulations in regions like southern Quebec, which is frequently influenced by convective storms in summer and large low-pressure systems in fall. These storms and low-pressure systems produce sometimes heavy rainfall in the summer and fall (Arkamose et al., 2011; Jeong & Sushama, 2018). Despite their advantages when compared with GCMs, 60 RCMs still tend to underestimate the intensity of heavy precipitation due to their inability to fully resolve small-scale processes (Kendon et al., 2012; Sunyer et al., 2012; Li et al., 2017), a limitation that is even more important in GCMs (Dai, 2006). Consequently, the use of RCM outputs for flood simulations can lead to an underestimation of flood magnitudes (Kay et al., 2006).Over the last decade, the emergence of Convection-Permitting Regional Climate Models (CP-RCMs), which operate at grid spacing of a few kilometers, has shown promise for improving the simulation of extreme precipitation events



(Kendon et al., 2012; Ban et al., 2014; Prein et al., 2015; Lucas-Picher et al., 2021). These models are capable of explicitly simulating deep convection, which is a key driver of intense precipitation (Berthou et al., 2020; Caillaud et al., 2021; Lucas-Picher et al., 2024). As a result, CP-RCMs have demonstrated improvements in simulating the frequency of extreme precipitation events, the diurnal cycle of rainfall, and hourly precipitation (Fosser et al., 2020; Vergara-Temprado et al., 2021). For example, Chan et al. (2014) found that a 1.5-km CP-RCM provided better agreement with observed extreme

hourly precipitation during the June–July–August (JJA) season compared to a coarser 12-km resolution RCM.

The ability of CP-RCMs to simulate heavy precipitation events, particularly on sub-daily timescales, makes them an important tool for flood simulations, especially in regions experiencing intense localized storms (Kay et al., 2015). Many studies have highlighted the added value of CP-RCMs in capturing local-scale phenomena such as convective cells and intense precipitation events, which occur at spatial resolutions finer than traditional RCMs (<5 km) (Kendon et al., 2012;

Prein et al., 2015; Coppola et al., 2020; Caillaud et al., 2021; Ban et al., 2021).

While RCMs have been widely used to drive hydrological models for flood simulations (e.g., Alfieri et al., 2015; Mendoza et al., 2016; Li et al., 2017; Qing and Wang, 2021), the added value of CP-RCMs, which offer more realistic simulations of extreme precipitation events, has recently started to be explored in hydrological-climate impact studies, particularly for flood simulations (Poncet et al. 2024). However, the effectiveness of CP-RCMs in simulating observed river discharge cycles has

been demonstrated in several regions. For instance, Mendoza et al. (2016) showed that CP-RCMs accurately simulated the annual cycle of river discharge in Colorado, while Lobligeois et al. (2014) found that CP-RCM precipitation can improve flow simulations in areas with highly variable precipitation. Rudd et al. (2020) and Quintero et al. (2022) reported that CP-RCMs outperformed traditional RCMs in simulating sub-daily precipitation and accurately reproducing river discharge when higher-resolution precipitation inputs (<5 km) is used to force hydrological models. Kay (2022) also noted that CP-RCMs

provided better performance than RCMs in simulating peak flows in reference period (1981-2000) across Great Britain. Schaller et al. (2020) further supported the use of higher-resolution climate model outputs for improving flood simulations in western Norway, emphasizing that CP-RCMs are more effective at capturing localized precipitation events than traditional RCMs. However, to our knowledge, no study has yet utilized CP-RCM outputs for flood simulations at an hourly time step in southern Quebec, Canada.

While the added value of CP-RCMs for flood simulations has been explored in other parts of the world, this remains an area of investigation for Quebec. The two primary causes of floods in southwestern Quebec are snowmelt (freshet) in the spring (Lucas-Picher et al., 2021; Riboust & Brissette, 2015; Oubennaceur et al., 2021) and heavy rainfall during the warmer seasons. The southern Quebec region is frequently affected by intense storms, which can produce heavy rainfall over short periods. Smaller basins are particularly vulnerable to such storms, which can lead to rapid and severe flooding. Therefore,

this study aims to assess the added value of CP-RCM for simulating floods in small basins, particularly those affected by intense rainfall during summer.

The Quebec government currently assess the impact of climate change on floods using a model chain that relies on climate scenarios based on GCMs and on RCMs and disseminates to numerous end-users the results in a Hydroclimatic Atlas





(MELCCFP 2022).  Numerous research projects like this one aims to improve future versions of this Atlas. This study

focuses on floods triggered by heavy rainfall from summer storms in southern Quebec, Canada. Convective storms can

generate intense rainfall over short periods (i.e., over few hours) across localized areas, typically spanning only a few

kilometers. These rare hourly rainfall events are particularly destructive in small basins, where a small volume of water is

sufficient to cause flooding. In contrast, larger basins in southern Quebec are primarily threatened by floods resulting from

snowmelt in spring, which falls outside the scope of this study. The primary objectives of our research are:

1.        Demonstrating the added value of CP-RCM in simulating extreme rainfall events during the summer and fall

(JJASON) period.

2.        Assessing whether this added value transfers to hydrological modeling, particularly for peak flow simulations.

The following sections present the methodology, results, and discussion, followed by the conclusion.

## 2 Data and study domain




**Figure 1: Domains of the CRCM6/GEM5 simulations: the green rectangle represents the CRCM6/GEM5-12km domain (a), while the blue rectangle represents the CRCM6/GEM5-2.5km domain (b). Selected basins of interest over**

**southern Quebec, Canada (c). Topography maps derived from CRCM6/GEM5-2.5km (d) and CRCM6/GEM5-12km (e). (The base map in Figure 1c was created using MATLAB® software by Esri. Copyright © Esri. All rights reserved. For more information about Esri® software, please visit www.esri.com)**





## 2.1 Study domain

This study is conducted over southern province of Quebec, Canada, more specifically over the south of the St. Lawrence River (Fig. 1c). This mid-latitude region has a temperate, humid climate with relatively homogeneous precipitation over the seasons occurring throughout the year. A total precipitation of around 1000 mm falls annually on average over this region, according to climate normal between 1991-2020 for Montreal (https://climate.weather.gc.ca/climate_normals), which is evenly distributed in 4 seasons. In the warm season, with larger amount of water vapor in atmosphere and high convective

activity, large rainfall events occur regularly. These events could trigger floods, particularly over smaller basins, that respond quickly to heavy rainfalls. Notably, this region includes major population centers such as Montreal and Quebec City, which receive around 500 mm of rainfall during the summer-fall period.

The basins were selected from the cartography of Quebec watersheds (Hydroclimatic Atlas; MELCCFP 2022) based on the availability of hourly river discharge measurements over many years. In this study, we focus on 11 basins, as shown on Fig.

1c, with details provided in Table 1). As shown in Fig. 1d, the topography derived from CRCM6/GEM5-2.5km indicates that the basins closer to the St. Lawrence River—particularly near Montreal—have lower elevations, whereas elevation increases further away from the river. Based on both basin size and elevation, the selected basins can be classified into five categories: A) 'near Montreal city' (Morpions ; Des Hurons ; L'Acadie ), B) 'near Sherbrooke city' (Eaton; Au Saumon), C) 'near Quebec city – smaller basins' (Boyer Sud; Bras d'Henri), D) 'near Quebec city – larger basins' (Bécancour; Etchemin;

Nicolet), E) northeast (Du Loup).

**Table 1. Information on river discharge stations, area, and mean elevation for each of the watersheds used in this study.**

| Category | Hydrometric station name | Provincial station number | Area (km$^2$) | Mean elevation (m) |
|---|---|---|---|---|
| C | Boyer Sud | 023002 | 61 | 173 |
| A | Morpions | 030423 | 94.1 | 65 |
| C | Bras d'Henri | 023432 | 154 | 161 |
| A | Des Hurons | 030415 | 308 | 43 |
| A | L'Acadie | 030421 | 367 | 50 |
| E | Du Loup | 022507 | 515 | 355 |
| B | Eaton | 030234 | 646 | 392 |
| B | Au Saumon | 030282 | 769 | 502 |
| D | Bécancour | 024003 | 914 | 346 |
| D | Etchemin | 023303 | 1152 | 384 |
| D | Nicolet | 030103 | 1550 | 237 |



## 2.2 Regional climate model and experimental setup

In this study, we use meteorological data from the sixth generation of the Canadian Regional Climate Model (CRCM6/GEM5), which is actively developed by a research group at the ESCER (Étude et simulation du climat à l'échelle régionale) center at UQAM (Université du Québec à Montréal) in collaboration with Environment and Climate Change Canada (ECCC). The CRCM6/GEM5 is based on the Global Environmental Multiscale model (GEM5) (McTaggart-Cowan et al., 2019), which is also employed by the Meteorological Service of Canada as its operational numerical weather prediction model.

In this study, we use two distinct configurations of the CRCM6/GEM5: one with a grid spacing of 12 km (CRCM6/GEM5-12km) and another with a grid spacing of 2.5 km (CRCM6/GEM5-2.5km). The CRCM6/GEM5-12km domain (Fig. 1a) is forced at its boundaries by the ERA5 reanalysis (Hersbach et al., 2020) 6-hourly 3D air temperature, relative humidity, geopotential height and winds (u and v vectors) developed at the European Centre for Medium-Range Weather Forecasts (ECMWF). Additionally, the CRCM6/GEM5-12km is also forced by ERA5 daily sea surface temperature (SST) and sea ice fraction.

The CRCM6/GEM5-2.5km domain centered over southern Quebec, Canada (Fig. 1b) is forced at its boundaries by hourly CRCM6/GEM5-12km outputs. The CRCM6/GEM5-12km uses the Kain-Fritsch deep convection parameterization scheme, while this deep convection parameterization in CRCM6/GEM5-2.5km is deactivated since it is explicitly simulated by the model (Roberge et al., 2024). Both CRCM6/GEM5 model configurations were run over the period 1999-2020. The years 1999 and 2000 considered as a spinup year were not used in this analysis.

## 2.3. Observed datasets

### 2.3.1 Meteorological data from weather stations:

Hourly 2-m temperature and precipitation from weather stations (Table 2) over southern Québec were extracted from the Environment and Climate Change Canada database (ECCC, 2025). In this study, weather station data are used to evaluate the CRCM6/GEM5 hourly precipitation. 26 weather stations located over southern Quebec were selected according to a criteria consisting to a data availability of at least 60% over the period (2014-2023). Although the temporal coverage of these weather stations (2014-2023) does not perfectly align with that of the climate simulations (2001-2020), they represent the best available observational data for this region.

River discharge from, discharge stations:

Observed 15-minute river discharges from the Ministère de l'Environnement, de la Lutte contre les Changements Climatiques, de la Faune et des Parcs (MELCCFP, 2025) are used to calibrate the hydrological model and as a reference for evaluating the simulated peak flows. The observed discharges are aggregated every hour using the mean to fit the temporal resolution of the hydrological model and CRCM6/GEM5 outputs.



### 2.3.2 Meteorological reanalyses:

To calibrate the hydrological model, we use the ERA5-Land (Muñoz-Sabater et al., 2021) 2-meter air temperature data. ERA5-Land provides high-resolution hourly meteorological data at a spatial resolution of 9 km. The precipitation data for calibrating the hydrological model comes from the Regional Deterministic Reforecast System version 2.1 (RDRSv2.1) (Gasset et al., 2021). Using a grid spacing of 10 km, RDRSV2.1 is a state-of-the-art reforecast system that provides high-quality precipitation estimates suitable for hydrological modeling. A comparison of the precipitation from RDRSv2.1 and

ERA5 with the weather stations indicated that RDRSv2.1 provided a better distribution of precipitation using intensity histograms and better hourly precipitation extremes using 99.99$^{th}$ percentile maps (not shown), explaining why we selected hourly precipitation from RDRSv2.1 to calibrate the hydrological model.

**Table 2. Summary of the observed and reanalysis datasets used in this study.**

| Dataset | Spatial resolution | Temporal resolution | Variable(s) | Period |
|---|---|---|---|---|
| Weather Stations (ECCC) | - | 1 h | Precipitation | 2014-2023 |
| Discharge Stations (MELCCFP) | - | 15-min | River discharge | 1997 – 2022 |
| Global Reanalysis ERA5-Land (ECMWF) | ~9 km | 1 h | 2-m air Temperature | 1999 – 2022 |
| Regional Reanalysis RDRSv2.1(ECCC) | ~10 km | 1 h | Precipitation | 2000 – 2018 |



# 3 Methodology

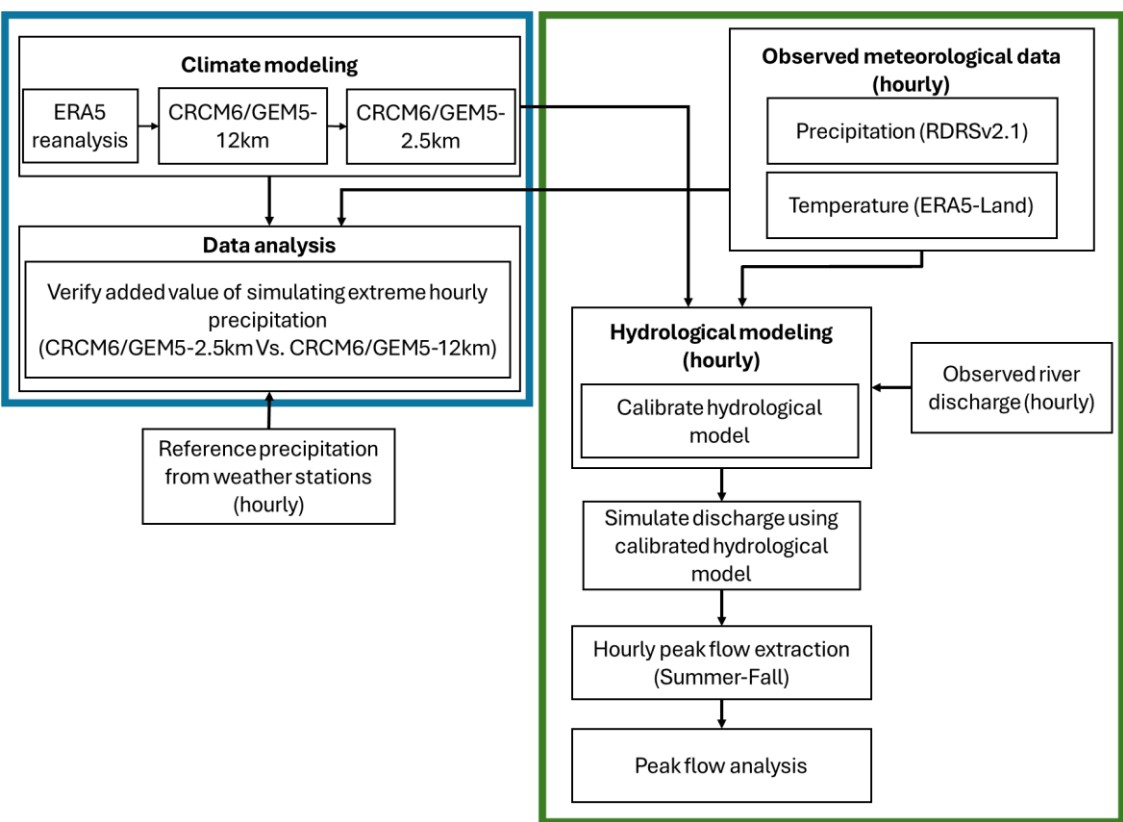


**Figure 2: Flowchart indicating the modeling chain used in the study.**

Figure 2 presents the modeling chain used in this study. It is divided into two sections, each of them addressing one of the objectives. In the left blue box, the added value of the simulated precipitation extremes from the CRCM6/GEM5-2.5km is assessed in a comparison with that from the CRCM6/GEM5-12km. Additionally, the different gridded precipitation datasets

are evaluated with the weather station data. The right box in green (Fig. 2) examines the potential benefits of using CRCM6/GEM5-2.5km for the simulation of floods.

## 3.1 CRCM6/GEM5-2.5km added value

To address the first objective of the study about the CRCM6/GEM5-2.5km added value, we compared hourly precipitation extremes from both CRCM6/GEM5 configurations with those of the weather stations. An accurate simulation of

precipitation extremes is crucial to simulated floods. The assessment of added value is done using two comparisons. The first comparison is done with intensity-frequency histogram, that illustrates the frequency of hourly precipitation of different magnitudes. Histograms are generated from hourly precipitation values greater than 0.1 mm/hr. The second comparison uses



the 99.99[th] percentile of each CRCM6/GEM5 configurations and the weather stations (percentiles calculated using the entire dataset). Given the hourly time step of the datasets, this corresponds the highest values obtained in average every 416 days.

## 3.2 Hydrological model

To simulate river discharge at an hourly time step, we used the GR5dt lumped hydrological model (Andrade et al., 2024), which is adapted from the GR4J and GR4H hydrological models (Génie Rural à 4 paramètres journalier; Perrin et al., 2003, Mathevet, 2005).

GR5dt has five parameters, which are watershed water exchange coefficient (mm), one-day maximal capacity of the routing reservoir (mm), unit hydrograph base time (days), percolation coefficient (dimensionless), production reservoir maximal capacity (mm). GR5dt has two reservoirs, production reservoir and routing reservoir. The model operates through a sequence of hydrological processes that transforms precipitation into river discharge.

Here is a simple explanation of how the hydrological model works (adapted from Perrin et al., 2003). The model begins by calculating net rainfall and potential evapotranspiration (PE). If rainfall exceeds PE, the excess contributes to the production reservoir (PR). Conversely, when PE surpasses rainfall, water is extracted from the PR. This step ensures that the production store is updated for subsequent hydrological processes. The production reservoir regulates infiltration (percolation leakage), which is computed as a power function of the storage level. This infiltration does not contribute significantly to the river discharge, but influences soil moisture dynamics.

The water exiting the PR, along with any excess rainfall not entering PR, is routed through two distinct pathways: 90% of the water is processed via the first unit hydrograph (UH1), followed by a non-linear routing store. 10% of the water is routed through a second unit hydrograph (UH2). Both hydrographs utilize the same time parameter to effectively distribute rainfall over multiple time steps. An exchange mechanism exists between groundwater and the routing store, affecting water availability in the system. The routing store is dynamically updated, considering the outflow from UH1 and the groundwater exchange process. The final outflow from the routing reservoir is combined with the output from UH2, forming the total river discharge at each time step. The combined outputs from the routing components represent the final river discharge generated by the model, effectively capturing the temporal distribution of runoff in the catchment.

To account for snow accumulation and melt processes, the Cemaneige snow module (Valéry et al., 2014) was coupled with the GR5dt hydrological model. Cemaneige has two parameters: snowmelt factor (mm/C), weighting coefficient of the thermal state of the snowpack (dimensionless). Cemaneige is a degree-day snow accounting model that improves river discharge simulation in snow-affected catchments by representing the dynamics of snow storage and melt. It divides precipitation into rain and snow based on air temperature and simulates snow accumulation and melt using a two-reservoir structure (Riboust et al., 2019). Snowmelt follows a temperature-index approach, where melt rate is proportional to temperature above a defined threshold. The model was applied with multiple elevation bands to better capture spatial variations in snow processes. By integrating Cemaneige with GR5dt, the hydrological model accounts for seasonal snow dynamics, improving the accuracy of river discharge simulations in cold and mountainous regions.





GR5dt requires two meteorological variables to simulate river discharges: hourly precipitation (taken from RDRSv2.1) and hourly 2-meter temperature (taken from ERA5-Land). The hourly 2-meter temperature is used to estimate potential evapotranspiration using the formula from Oudin et al. (2005). Additionally, a simple linear temperature-based method (Kienzle, 2008) is applied to separate rainfall from snowfall in the precipitation data.

The interpolation of gridded data was performed using the Thiessen polygon method (Brassel and Reif, 1979). For basins smaller than 100 km², grid points located within the basin's boundary were weighted based on Thiessen polygons. A single representative value for each basin was then computed by applying the corresponding weights to the grid points within the basin. For basins larger than 100 km², a single representative value was obtained by assigning equal weight to all grid points located within the basin's boundaries.

**3.2.1 Hydrological model calibration**

GR5dt is calibrated by identifying the 7 parameters that provide the best performance (based on the performance metric Kling-Gupta Efficiency (KGE; Gupta et al., 2009; Kling et al., 2012)) using the Shuffled Complex Evolution-University of Arizona (SCE-UA) algorithm (Duan et al., 1994). Here, the GR5dt simulates river discharge over the entire time series (2001–2018) using hourly precipitation from RDRSv2.1 and hourly 2-meter temperature from ERA5-Land. The simulations

are initialized with parameter sets using initial seeds and half-empty reservoirs. Once GR5dt generates a continuous river discharge time series, the performance metric KGE is calculated comparing the simulated and measured river discharge. KGE evaluates GR5dt performance based on three components: bias, correlation, and variability of the simulated river discharge compared to that observed.

$$KGE = 1 - \sqrt{(r-1)^2 + (\beta-1)^2 + (\gamma-1)^2} \tag{1}$$

$$\beta = \frac{\mu_s}{\mu_o} \qquad \gamma = \frac{\sigma_s/\mu_s}{\sigma_o/\mu_o} \tag{2}$$

Where r is the Pearson correlation coefficient between the simulated (s) and observed (o) river discharges, $\beta$ is the bias ratio, $\gamma$ is the variability ratio, $\mu$ is the mean river discharge, and $\sigma$ is the standard deviation.

A KGE of 1 indicates a perfect match between observed and simulated river discharges. A KGE value of around -0.41 corresponds to the performance equivalent of using the mean simulated flow. A KGE lower than -0.41 signifies that model

performance is worse than using the mean flow as basic model (Knoben et al., 2019). The KGE is then used as the objective function in the optimization algorithm, defined as: Objective function = 1 - KGE. The SCE-UA algorithm is employed to minimize this objective function, aiming to achieve the lowest possible value. The optimization process is repeated 5,000 times, a number chosen to balance computational efficiency with the quality of the resulting objective function. At the end of the calibration iterations, the parameter set that produces the lowest objective function value is selected for the next step.

Once the hydrological model has been calibrated with observed data, it is ready to be run using the climate model output. Due to the lack of high-quality high-resolution observed gridded precipitation and temperature datasets, we decided to not perform bias correction to avoid the degradation of the CRCM6/GEM5 meteorological variables.





### 3.3 Assessing the benefits of CRCM6/GEM5-2.5km over CRCM6/GEM5-12km for simulating peak flows

In this section, we evaluate which CRCM6/GEM5 configuration produces the best peak flows. We simulate hourly river

discharge, using the GR5dt hydrological model forced by climate simulations from both CRCM6/GEM5 configurations (2.5 and 12 km).

### 3.3.1 Hourly peak flow analysis

Once GR5dt is calibrated, three hydrological simulations over the period 2001-2018 are performed with different forcings.

- o Simulation 1: Precipitation from RDRSv2.1 and 2-meter temperature from ERA5-Land.
- o Simulation 2: Precipitation and 2-meter temperature from CRCM6/GEM5-12km.
- o Simulation 3: Precipitation and 2-meter temperature from CRCM6/GEM5-2.5km.

To assess the ability of GR5dt to produce floods in the 3 simulations, we selected the peak flows, consisting to the highest river discharge value between June 1 and November 30 of each year. This period was chosen because deep convection is an important factor of extreme precipitation events during this time period, and it is explicitly simulated by CRCM6/GEM5-

2.5km, which increases the likelihood of highlighting the benefits of using this high-resolution model. To assess the performance of the simulations that span 18 years, 18 peak flows from the observed river discharge records were also selected. Among the 11 basins, four of them ('Bécancour,' 'DesHurons,' 'Etchemin,' and 'L'Acadie') have 17 peak flows instead of 18 due to missing data in the observed timeseries. For these basins, the lowest simulated peak flows were removed for the analysis.

Three metrics are considered to assess the ability of the 3 simulations to reproduce floods. First, we calculate the normalized Root Mean Square Error (nRMSE) between simulated peak flows (Peak_Qsim) and observed peak flows (Peak_Qobs). RMSE represents the average squared difference between the i-th sorted 18 peaks from the simulated and observed hourly river discharges. The RMSE is then normalized by the average of the 18 peak flows from the corresponding dataset. An nRMSE of zero indicates that the simulated peaks have the same magnitude as those observed. nRMSE = $[0, \infty)$.

$$nRMSE = \frac{\sqrt{\frac{\sum_{i=1}^{N}(Peak\_Qsim_i - Peak\_Qobs_i)^2}{N}}}{(\sum_{i=1}^{N} Peak_{Qobs_i})/N} \qquad (3)$$

where N is 18, and i is corresponding to the rank of the peak among total of N peaks.

The nRMSE indicates which simulation is more accurate in reproducing the observed hourly peak flows during the summer-fall seasons (2001–2018). Using nRMSE introduces a deterministic aspect to the comparison by reducing the assessment to a single value, simplifying the distinction between different cases. This straightforward approach assesses the performance of

the simulated peak flows of each simulation using a single value.

Another metrics consists of the biases of each sorted simulated peak flows.

$$bias_i = \frac{Peak\_Qsim_i - Peak\_Qobs_i}{Peak\_Qobs_i} \qquad (4)$$





Where i corresponds to the rank of the peak flows for each simulation and the observation.

These results are presented using boxplots, which provide a general overview of bias dispersion for each dataset. Each
boxplot corresponds to one dataset used to force the hydrological model and contains 18 annual peak bias samples. The third
comparison focuses specifically on the summer-fall peaks to determine which simulations show better agreement with
observed peak flows. Since larger peak flows are more destructive, this analysis provides valuable insight. To evaluate the
statistical distribution of peak flows, we extracted normalized seasonal peak flows, defined as the maximum hourly river
discharge values occurring between June and November each year, from the simulated and/or observed streamflow time
series for each basin. The peak flows were normalized by dividing the river discharge values by the corresponding basin
area, to allow for consistent comparisons across basins of different sizes. These normalized values were then used to
construct empirical Cumulative Distribution Functions (CDFs), which represent the probability that a given peak river
discharge per unit area is less than or equal to a specific value. This method enables direct comparison of the distribution and
magnitude of extreme events across different datasets. The CDFs were plotted to visually assess the frequency and intensity
of high-flow events and to evaluate the model's ability to capture the upper tail of the flow distribution, which is essential for
flood risk assessment.

## 4 Results

### 4.1 Assessing the added value of CRCM6/GEM5-2.5 in simulating extreme rainfall

Two analyses were conducted to determine if the CRCM6/GEM5-2.5km provides a more realistic representation of extreme
hourly precipitation than CRCM6/GEM5-12km. Figure 3 shows the intensity-frequency histogram for one of the 26 weather
stations (for the histograms of the remaining stations see Fig. S1- S25 in the Supplement). It is worth noting that, due to
differences in data availability periods between weather stations and other sources (see Table 2), a 10-year period was
selected to produce the results shown in Fig. 3. This choice ensures consistency across datasets, as the same time series
length was used when calculating the intensity–frequency histograms. For other datasets, the most recent 10-year period
available was selected.

The same analysis is performed with the precipitation from RDRSv2.1 to determine the ability of the dataset used for the
calibration of the hydrological model in simulating precipitation extremes. The histograms are computed using only hourly
precipitation values greater than 0.1 mm hr$^{-1}$ (so-called wet hours (Pichelli et al., 2021)). Both CRCM6/GEM5
configurations show a good agreement with the weather stations in the fall (Fig. 3b- 3d). However, differences in
precipitation intensity become more evident in the summer (Fig. 3a- 3c). CRCM6/GEM5-2.5km produces stronger
precipitation extremes than CRCM6/GEM5-12km, which are comparable to those observed. The percentage of wet hours is
provided in each figure's inset box, showing that CRCM6/GEM5-12km generally simulates more wet hours than
CRCM6/GEM5-2.5km.



The most noticeable difference among the four datasets presented in Fig. 3 is the distribution of yellow bins, which represent
the CRCM6/GEM5-12km dataset. These bins appear predominantly on the left side of the histograms in Fig. 3c- 3d,
particularly during the summer season. This pattern indicates that the CRCM6/GEM5-12km dataset tends to underestimate
heavy precipitation events compared to the other datasets. Such an underestimation could be attributed to the model's
resolution, parameterization schemes, and inherent biases in representing intense rainfall events. To assess the accuracy of
extreme precipitation values, the 99.99$^{th}$ percentile threshold (i.e. considering all data for calculating the percentile) is
indicated with a solid vertical line in Fig. 3. The red lines (from CRCM6/GEM5-2.5km) are closer to the black lines (from
the weather station), indicating a better agreement with the observed extreme precipitation. In contrast, the yellow lines
(CRCM6/GEM5-12km) are smaller from those of the weather stations, highlighting an underestimation of extreme rainfall
events. Although the percentage of wet hours is higher in the coarser-resolution climate model, the higher-resolution
CRCM6/GEM5-2.5km shows a better agreement with the weather station data for the extreme precipitation events. The
strong agreement between the CRCM6/GEM5-2.5km and the weather station for the intense precipitation increases the
confidence in the reliability of the higher-resolution model. Simulating explicitly the deep convection, the finer
CRCM6/GEM5-2.5km simulates better precipitation extremes, often associated to convective rainfall events, whereas for the
coarser CRCM6/GEM5-12km, the precipitation extremes seem to be smooth out due to the use of a deep convection
parameterization and coarser resolution.

There is a strong agreement between RDRSv2.1 with the weather station data. The RDRSv2.1 closely matches the frequency
of heavy precipitation events observed by the weather station and provides similar values for extreme precipitation,
particularly at the 99.99$^{th}$ percentile. This match suggests that RDRSv2.1 is a more reliable dataset for representing extreme
rainfall events compared to other gridded datasets. Interestingly, the proportion of wet hours in RDRSv2.1 and
CRCM6/GEM5-2.5km is closer to the weather station data compared to CRCM6/GEM5-12km.

While Fig. 3 presents results for a single weather station, similar patterns are observed across other locations, as provided in
the Supplement (see Fig. S2- S25 in the Supplement). The consistency of these results suggests that CRCM6/GEM5-2.5km
is more reliable for capturing extreme precipitation than CRCM6/GEM5-12km. The underestimation of CRCM6/GEM5-
12km is particularly pronounced in summer, likely due to its inability to fully resolve deep convection, which is the
dominant precipitation mechanism in the warm months.

Overall, the results indicate that CRCM6/GEM5-2.5km provides a more realistic representation of both general precipitation
and extreme values. This improvement is especially evident in summer, when localized, high-intensity rainfall events play a
more significant role in shaping the precipitation distribution.



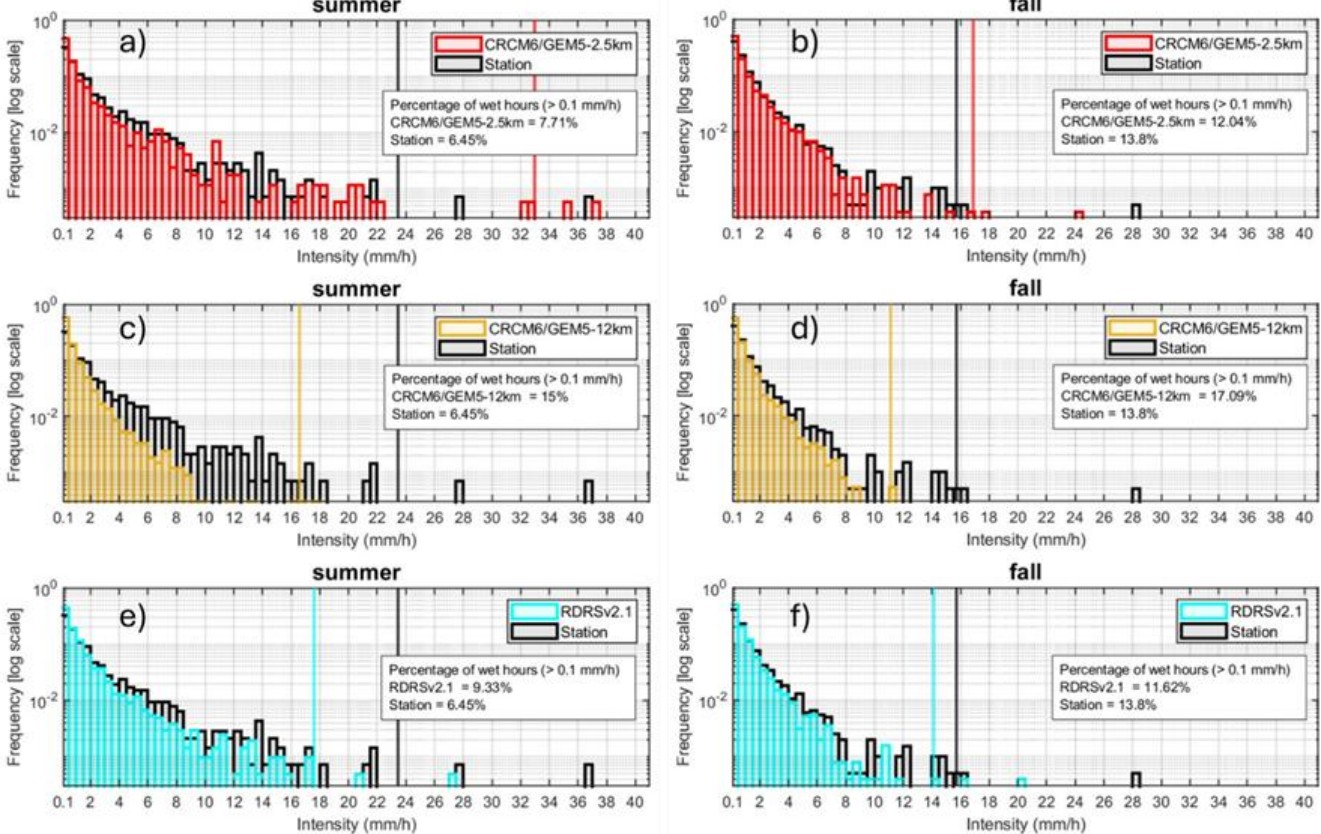

**Figure 3. Intensity-frequency distribution of hourly precipitation (>0.1 mm per hour) over a 10-year period for the weather station of the airport Montréal-Trudeau. Black bins represent weather station data (2014–2023), yellow bins correspond to CRCM6/GEM5-12km (2011–2020), red bins correspond to CRCM6/GEM5-2.5km (2011–2020), and cyan bins correspond to RDRSv2.1 (2009–2018). The vertical line represents the 99.99$^{th}$ percentile (based on all values) for each dataset. Left-side panels correspond to summer seasons (a, c, e), and right-side panels correspond to fall seasons (b, d, f). The bin size is 0.5 mm.**

The second assessment employs a comprehensive approach, conveying a substantial amount of information through maps. Figure 4 presents six maps of the study area, four maps for different climate model resolutions: CRCM6/GEM5-2.5km (Fig. 4a- 4b) and CRCM6/GEM5-12km (Fig. 4c- 4d), and two maps for gridded reanalysis, RDRSv2.1 (Fig. 4e- 4f). These maps correspond to the 99.99$^{th}$ percentile of the hourly precipitation for each grid cell. Overlaid on these maps are square markers indicating the 99.99$^{th}$ percentile values derived from the weather stations. The agreement between the gridded datasets and the weather station can be visually assessed by comparing the colors of the grid cells with those of the markers; a closer match indicates better alignment with observed data from weather stations.

For all datasets, extreme precipitation values are generally higher in summer than in fall. This pattern is also evident for weather stations, where most stations—except for two stations, there are 26 stations in total—report higher extreme values





during summer. This aligns with established climatological understanding, as higher temperatures in summer favor stronger convective activity, leading to more intense precipitation events.

The higher spatial resolution of CRCM6/GEM5-2.5km allows for an explicit simulation of deep convection. Moreover, the finer resolution enhances the ability of the CRCM6/GEM5 to simulate orographic precipitation. The ability of high-
resolution models to resolve fine-scale features is particularly important when simulating extreme precipitation, as such events often occur over small spatial extents. For instance, a localized convective storm can result in intense rainfall over just a few kilometers, while areas only slightly further away may receive significantly less precipitation. These sharp spatial contrasts, often influenced by orographic effects and mesoscale meteorological processes (Houze Jr, 2012), are not well represented in the coarser resolution model i.e. CRCM6/GEM5-12km. This discrepancy is evident in Fig. 4, where the
precipitation intensity seen in CRCM6/GEM5-2.5km contrast sharply with the more generalized patterns in CRCM6/GEM5-12km.

Furthermore, CRCM6/GEM5-12km systematically underestimates the 99.99th percentile values for both seasons. This underestimation is expected, as coarse-resolution models struggle to resolve localized extreme precipitation events that develop over small areas (Ban et al 2014; Prein et al., 2020; Vogel et al., 2021). The inability of CRCM6/GEM5-12km to
capture these localized extremes results in a systematic underestimation of extreme precipitation at sub-daily time scales. In contrast, the CRCM6/GEM5-2.5km simulations exhibit better agreement with weather station data, demonstrating the advantages of higher spatial resolution in capturing extreme hourly rainfall (Fig. 4). Despite some discrepancies between weather stations and the climate models—partly due to differences in the observational record and model simulations periods—the overall pattern suggests that CRCM6/GEM5-2.5km provides a more reliable representation of extreme
precipitation events than CRCM6/GEM5-12km. This improved agreement is particularly crucial for the simulation of summer extremes, where small-scale convective storms dominate precipitation processes. As for Fig. 3, RDRSv2.1 is showing a good agreement with weather stations, comparing the colors of the squares in the map (Fig. 4e- 4f) with the surroundings grid colors.







**Figure 4: Maps of 99.99th percentile of hourly precipitation (based on all values). Each square represents the 99.99th percentile from a weather station (2014–2023), climate model outputs (CRCM6/GEM5-12km & CRCM6/GEM5-2.5km) correspond to the period 2011–2020, and the RDRSv2.1 reanalysis corresponds to the period 2009-2018. The left-side maps (a, c, e) show summer results (June 1–August 31), while the right-side maps (b, d, f) show fall results (September 1–November 30). Basin contours are also shown (refer to Fig. 1c for basin names).**

**4.2 Assessing the added value of using higher resolution climate model outputs for simulating hourly floods**

Figure 5 presents the KGE values for using the three datasets, assessing their ability to simulate river discharge during the summer and fall seasons. Each data point represents the KGE obtained by comparing simulated river discharge time series against observed values. Simulated river discharges are generated using the same parameter set from the previously calibrated hydrological model and are forced with different datasets.

Since the hydrological model was calibrated using reanalysis datasets (RDRSv2.1 for precipitation and ERA5-Land for temperature), RDRSv2.1 produces the highest KGE values. The strong agreement between RDRSv2.1 and observed river discharge indicates that the simulated river discharge is a reliable reference for model evaluation.

The KGE values for CRCM6/GEM5-12km (yellow squares) and CRCM6/GEM5-2.5km (red circles) are notably lower than RDRSv2.1. This lower performance is expected because regional climate models are not designed to replicate the sequence of observed events since they are forced only at their lateral boundaries by ERA5 and they are affected by the chaotic nature of the atmosphere (Lucas-Picher et al. 2008).

The results suggest a slight increase in KGE values as basin size increases (moving from left to right). This trend may be attributed to the fact that, for larger basins, errors from various sources—such as rainfall estimation and model parameterization—tend to offset each other, resulting in a more stable hydrologic response. In contrast, smaller basins are more sensitive to uncertainties, as localized errors in precipitation inputs have a more pronounced effect on the simulated river discharges.



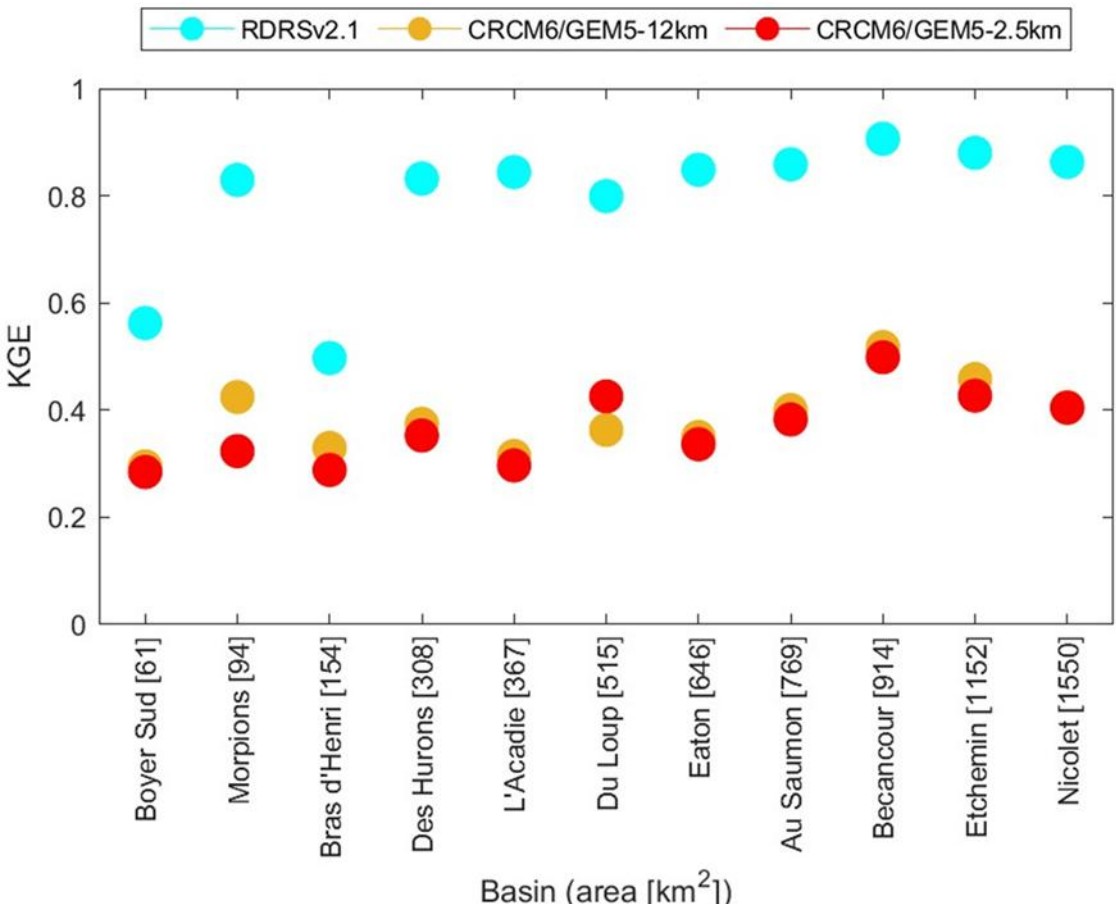

**Figure 5: Hydrological model performance (KGE) based on running the calibrated model with the reanalysis (cyan – RDRSv2.1) and climate model outputs (yellow - CRCM6/GEM5-2.5km; red - CRCM6/GEM5-12km).**


Figure 6 presents peak river discharges for each calendar year between June and November (summer–fall season). For each year, the highest hourly river discharge value is selected as the annual peak flow. This results in 18 peak flows from four datasets: observations (measured), and simulated using RDRSv2.1, CRCM6/GEM5-12km, and CRCM6/GEM5-2.5km. The river discharges are normalized by the basins' area (m³/s/km²), allowing a direct comparison of peak flow intensity across

basins of different sizes. The y-axis lists the basins from the smallest (bottom) to largest (top). Such organization highlights the influence of basin size on peak flow dynamics.

Normalized peak flows are generally larger for smaller basins. The smallest basin (basins in category C), Boyer Sud (61 km²) and third smallest one, Bras d'Henri (154 km²), exhibit the most extreme peak flows, reinforcing the well-established hydrological principle that smaller basins respond more rapidly to intense rainfall (Blöschl & Sivapalan, 1995), resulting in

sharper and more extreme river discharges. The similarity in peak flows for these two basins could be attributed to their geographic proximity and similar hydrological characteristics.





The hydrological model simulations with the three datasets underestimate peak flows for the two most intense peak flows compared to observations. This underestimation is not limited to peak simulations using climate model outputs (yellow: CRCM6/GEM5-12km and red: CRCM6/GEM5-2.5km), but also those used reanalysis data (cyan: RDRSv2.1) as well, which highlights the challenging task of simulating strong floods of smaller basins. However, for the two basins with the highest observed peak flows, the largest simulated peaks using CRCM6/GEM5 at 2.5 km resolution are closer to the observations. This suggests that finer spatial resolution could enhance the climate model's ability to capture localized extreme rainfall, which directly influences peak river discharges. This highlights the advantage of using CRCM6/GEM5-2.5km data for flood simulations compared to CRCM6/GEM5-12km, although cautions need to be taken for generalization these findings.

As basin size increases (moving upward on the y-axis of Fig. 6), normalized peak flows become generally weaker and more clustered. This pattern aligns with hydrological expectations, as larger basins tend to buffer extreme rainfall events due to slower response times and more distributed runoff. Additionally, errors in precipitation estimation have a smaller impact on larger basins due to spatial averaging effects, which reduce variability in simulated peak flows.

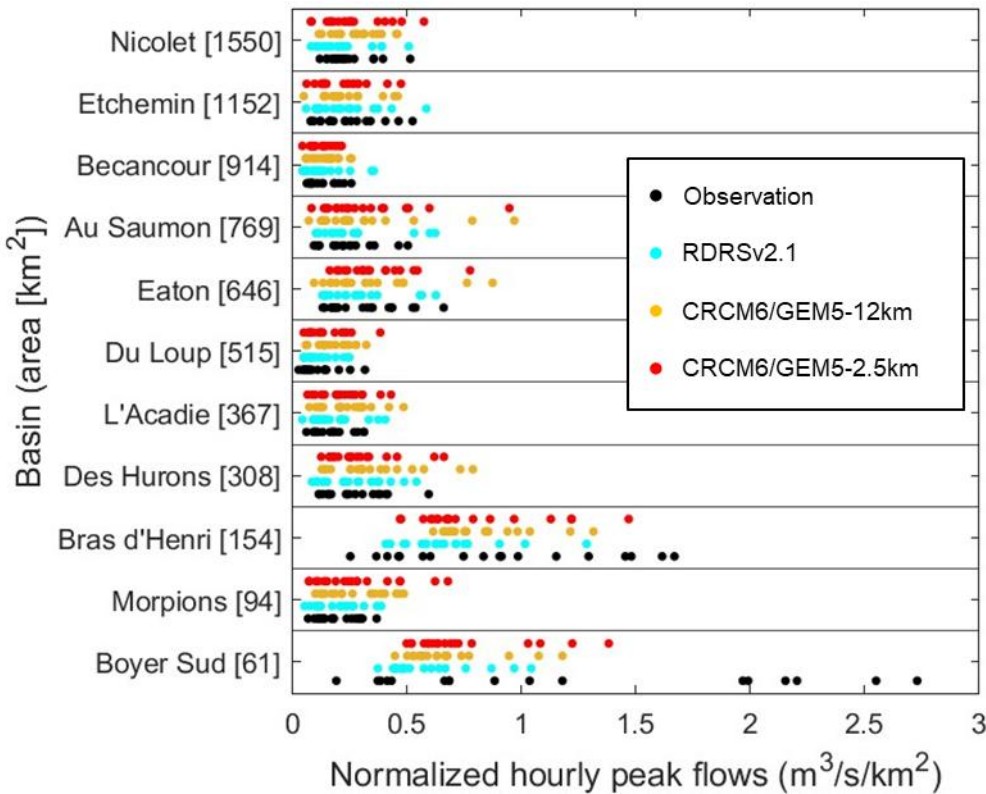

**Figure 6: Comparison of observed and simulated hourly peak flows normalized with the basin area (m³/s/km²) for the summer-fall seasons (2001-2018) across 11 basins of varying sizes (small (bottom) to large (top). Each circle**





**represents a peak flow event, with black indicating observations, cyan RDRSv2.1, yellow CRCM6/GEM5-12km, and red CRCM6/GEM5-2.5km.**


Figure 7 presents boxplots of biases of sorted (lowest to largest) simulated peak flow for the three different datasets: two climate models (CRCM6/GEM5-2.5km and CRCM6/GEM5-12km) and reanalysis (RDRSv2.1). The performance of the simulated peak flows is evaluated based on two criteria: 1) Boxplot size (dispersion of biases): smaller boxes indicate lower variability in biases, meaning the simulation produces consistent results, and 2) Symmetry around zero bias (vertical dashed line): a boxplot centered around the zero-bias line suggests that the simulation has little systematic overestimation or underestimation of peak flows. Based on these criteria, the reanalysis dataset (RDRSv2.1) exhibits the best agreement with observed peak flows compared to the climate model outputs. This result is expected since the hydrological model was calibrated using reanalysis data, making it inherently more aligned with observed river discharges. In contrast, the simulated peak flows using the climate model datasets generally are generally overestimated, as seen in the upward shift of their boxplots. However, it is important to acknowledge that this evaluation is based on an 18-year period, whereas climate studies typically use a 30-year period. A longer evaluation period would provide a more robust assessment of model performance in capturing peak flows.

Among the simulated peak flows using the two climate model simulations, the ones using CRCM6/GEM5-2.5km generally performs better than those of CRCM6/GEM5-12km. This can be seen through closer to 0 bias, and relatively smaller and more symmetric boxplots of CRCM6/GEM5-2.5km compared to CRCM6/GEM5-12km, indicating reduced bias variability and improved accuracy in simulating peak flows. The added value of the higher-resolution model (CRCM6/GEM5-2.5km) in simulating peak flows is particularly noticeable for the larger basins and those experiencing stronger peak flows.




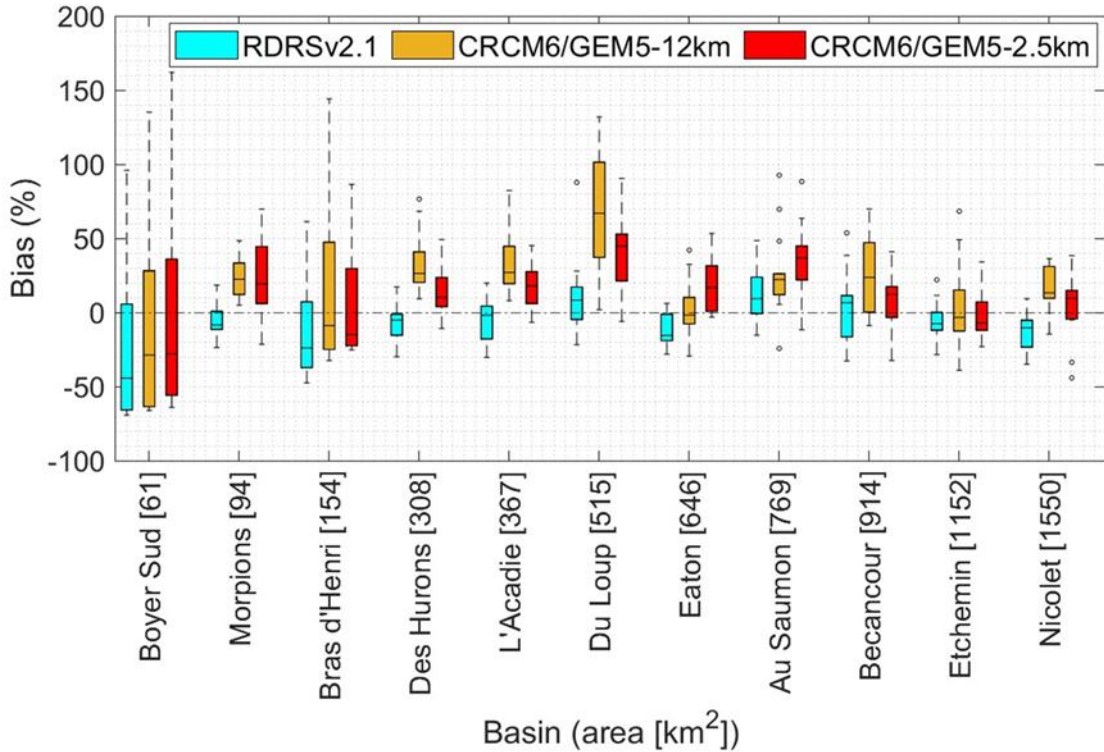

**Figure 7: Bias (%) of sorted simulated peak flows (lowest to largest) across different basins. Cyan represents**
**reanalysis (RDRSv2.1), red corresponds to CRCM6/GEM5-2.5km, and yellow denotes CRCM6/GEM5-12km. The**
**horizontal dashed line indicates zero bias.**

**Table 3. nRMSE values for peak flow simulations from CRCM6/GEM5-2.5km, CRCM6/GEM5-12km, and**
**RDRSv2.1 across different basins. Bold font indicates the lowest nRMSE value for each row (basin).**

| nRMSE | | Simulations | | |
|---|---|---|---|---|
| | | CRCM6/GEM5-2.5km | CRCM6/GEM5-12km | RDRSv2.1 |
| Basin [area in square kilometers] | Boyer Sud [61] | **0.68** | 0.74 | 0.77 |
| | Morpions [94] | 0.50 | 0.34 | **0.09** |
| | Bras d'Henri [154] | **0.24** | 0.31 | 0.38 |
| | Des Hurons [308] | 0.22 | 0.44 | **0.11** |
| | L'Acadie [367] | 0.21 | 0.35 | **0.17** |
| | Du Loup [515] | 0.35 | 0.55 | **0.19** |
| | Eaton [646] | 0.17 | 0.25 | **0.16** |
| | Au Saumon [769] | 0.58 | 0.62 | **0.27** |





| | Becancour [914] | **0.16** | 0.28 | 0.33 |
|---|---|---|---|---|
| | Etchemin [1152] | 0.15 | 0.16 | **0.12** |
| | Nicolet [1550] | 0.20 | 0.21 | **0.13** |

To determine whether CRCM6/GEM5-2.5km or CRCM6/GEM5-12km is more suitable for flood simulation, we assess their performance using a single evaluation metric, as shown in Table 3. This table presents the results using both climate model outputs across different basins, making it straightforward to answer the question: 'Which climate model allows the best simulated peak flow magnitudes?'

The results indicate that CRCM6/GEM5-2.5km generally leads to peak flows that are closer to those observed. For 10 out of

11 basins, CRCM6/GEM5-2.5km shows lower nRMSE values compared to CRCM6/GEM5-12km. The only exception is the Morpions basin, where CRCM6/GEM5-12km performs better (i.e., has a lower nRMSE). Notably, the Morpions basin experiences smaller peak flows (as seen in Fig. 6), suggesting that the added value of CRCM6/GEM5-2.5km is more pronounced for basins experiencing larger peak flows. This aligns with the fact that CRCM6/GEM5-2.5km better simulates extreme precipitation. Thus, when extreme precipitation occurs over a highly reactive basin, higher-resolution climate model

outputs are more beneficial in simulating peak flows. However, for basins that experience weaker peak flows, the added value of higher-resolution outputs may be smaller.

Interestingly, for two of the smallest basins—Boyer Sud (the smallest) and Bras d'Henri (the third smallest)—CRCM6/GEM5-2.5km yields lower nRMSE values than RDRSv2.1. This highlights the added value of CRCM6/GEM5-2.5km in simulating peak flow magnitudes, even when compared to reanalysis data (RDRSv2.1). Biases decrease as basin

size increases due to the smoothing effect of various sources of uncertainty. Larger basins tend to exhibit less biased peak flows. This trend related to basin size becomes more evident when comparing results from basins with similar geographic characteristics (Fig. 1) and peak flow dispersion (Fig. 6). For example, among the smallest basins, 'Boyer Sud' and 'Bras d'Henri' exhibit similar bias dispersion and interquartile ranges. Additionally, as basin size increases, biases become more symmetric around zero with a smaller range of dispersion.

Figure 8 shows cumulative distribution functions of sorted (smallest to largest) normalized peak flows, comparing the simulated ones with the observed ones. Climate models are not designed to reproduce the precise sequence of events, but rather to capture the overall climate characteristics. In other words, simulated peak flows are not necessarily happening on the same year of those observed. Therefore, the goal here is to assess how well the climate model outputs can reproduce the distribution of peak flows.

Each of the 11 subplots consists to a different basin using peaks flows of hourly river discharges in summer-fall period of each calendar year. Each point on the plot represents a ranked normalized peak flow event. Specifically, a given point i corresponds to a cumulative probability $Y_i$ on the vertical axis and a normalized peak flow value $X_i$ on the horizontal axis. $Y_i$ indicates the fraction of peak flows of a dataset, either observed or simulated, that their magnitudes are lower than $X_i$. Ideally, the simulations should closely overlap the observations. The closer the model follows the observed CDF, the





better it represents the magnitude of the observed peak flows. The CRCM6/GEM5-2.5km simulation shows smaller differences from the observations compared to the CRCM6/GEM5-12km simulation, suggesting a better representation of peak flow distributions. The results suggest that higher resolution (2.5 km) generally improves simulation accuracy, though some biases remain. Some basins show larger discrepancies between simulations and observations, suggesting spatial variability in hydrological model performance across different hydrological conditions.

The comparison of CDFs across 11 basins reveals heterogeneous behavior in the simulation of peak flows depending on basin size, data source, and model resolution. In smaller basins, model performance varies notably: for example, in Boyer Sud, both CRCM6/GEM5 simulations (2.5 km and 12 km) tend to underestimate peak flows, while in Morpions, both resolutions overestimate them. This contrast highlights the complexity of representing localized hydrological responses in small catchments. In intermediate-size basins such as Des Hurons and L'Acadie, the CRCM6/GEM5-12km model often

simulates higher peak flows than the CRCM6/GEM5-2.5km version, likely due to the spatial aggregation of intense precipitation across larger grid cells in the coarser model, leading to greater total runoff volumes. In larger basins like Etchemin and Nicolet, the differences between resolutions diminish, and the simulations from all sources tend to converge toward observed values. The RDRSv2.1 reanalysis product generally falls between observations and CRCM6/GEM5 simulations and shows better agreement with observations in some basins, such as Du Loup and Nicolet. These results

emphasize the importance of considering basin size and model resolution when evaluating simulated peak flows for flood risk assessments.





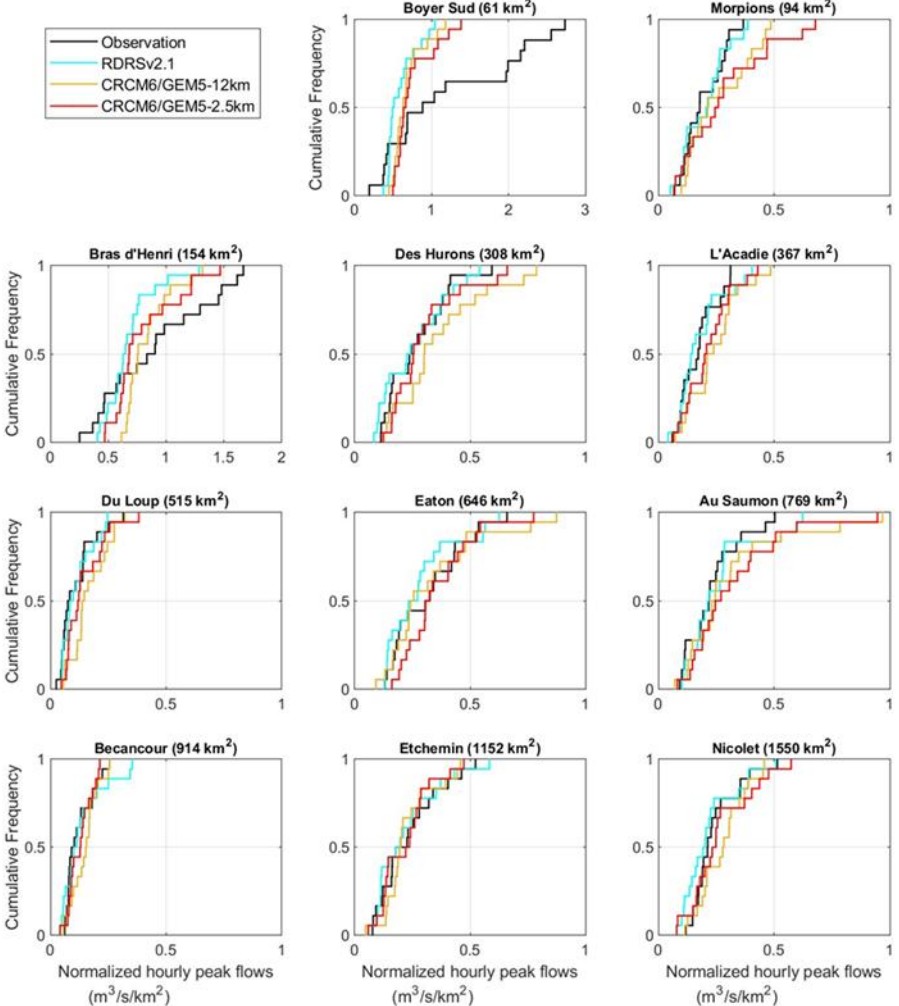

**Figure 8: Cumulative distribution functions (CDFs) of sorted (smallest to largest) normalized hourly peak flows across basins. Black lines indicate the observations, red lines indicate CRCM6/GEM5-2.5km and yellow lines indicate**
**CRCM6/GEM5-12km. Basins are sorted from smallest at top left to biggest at bottom right. To improve plot readability, the two basins (Boyer Sud; Bras d'Henri) with the strongest peak flows are shown with a different x-axis scale from the others.**

Figure 9 presents the biases in simulated peak flows using both climate model outputs compared with respect to the observed peak flows. The dashed vertical line represents zero bias, meaning that the closer the lines are to this reference, the less
biased the simulations are in representing peak flows. For most basins, the red lines (CRCM6/GEM5-2.5km) are closer to the dashed vertical line compared to the yellow lines (CRCM6/GEM5-12km), indicating lower biases.







**Figure 9: Ranked peak flows versus biases. The values are sorted from the largest at the top of each plot to the smallest at the bottom. The vertical dashed lines correspond to zero deviations from observed peak flows. The data represents peak flows from 2001 to 2018 during the summer and fall seasons. Red lines correspond to CRCM6/GEM5-2.5km, while yellow lines correspond to CRCM6/GEM5-12km. Basins are sorted from the smallest at top left to biggest at lower right.**





## 5 Discussion

### 5.1 Added value of using higher resolution climate model

This study first evaluated the added value of high-resolution climate model outputs (CRCM6/GEM5-2.5km) in representing extreme rainfall. The results demonstrate that CRCM6/GEM5-2.5km offers a more accurate representation extreme precipitation events compared to CRCM6/GEM5-12km. According to the Clausius-Clapeyron relationship, the atmosphere's capacity to hold water increases by approximately 7% per degree Celsius of warming (Trenberth et al., 2003; Martinkova & Kysely, 2020), which enhances cloud formation and ultimately leads to significant rainfall. In this study, we focused on the warm season—summer and fall—since higher temperatures during this time of the year result in greater water vapor availability, creating favorable conditions for storms. Consequently, warmer conditions have a higher potential for intense precipitation events (Meyer et al., 2022). Additionally, extreme rainfall can occur on small spatial scales, such as localized heavy precipitation from convective storms. Previous studies have shown that coarse-resolution climate models tend to underestimate heavy precipitation at sub-daily timescales (Kendon et al., 2014).

The comparison of precipitation intensity distributions (Fig. 3) highlights a key limitation of climate models with coarser resolution: the tendency to underestimate extreme rainfall events. This discrepancy arises from the inability of CRCM6/GEM5-12km to resolve fine-scale convective processes, which are critical for accurately simulating short-duration, high-intensity rainfall. The superior performance of CRCM6/GEM5-2.5km in capturing extreme precipitation aligns with findings from previous studies (e.g., Ban et al., 2014; Prein et al., 2020), which have shown that finer spatial resolution enhances the representation of convective systems. The spatial representation of extreme precipitation (Fig. 4) further reinforces these findings. The finer grid of CRCM6/GEM5-2.5km captures localized variations in extreme precipitation better than CRCM6/GEM5-12km. These differences are particularly pronounced in the summer, when convective storms dominate precipitation patterns. The improved performance of CRCM6/GEM5-2.5km in capturing high-intensity rainfall events suggests that high-resolution climate models offer added value for applications requiring accurate extreme precipitation estimates, such as flood risk assessments.

### 5.2 Transferability of skills from extreme precipitation simulation to peak flow simulation

Showing that climate model outputs at 2.5 km horizontal resolution are better than the coarser resolution simulating extreme precipitation is promising for better simulation of extreme floods. However, improvements in precipitation extreme do not consistently translate into better simulated river discharges. While higher-resolution climate models and improved precipitation can enhance the representation of extreme rainfall events, uncertainties in hydrological modeling, parameter calibration, and catchment-specific characteristics can limit the direct benefits of these improvements (Kay et al., 2015; Kay, 2022; Reszler et al., 2018).

Hydrological models are sensitive to various factors, including soil moisture dynamics, land surface processes, and runoff generation mechanisms, which may not always respond linearly to more accurate precipitation inputs (Zehe et al., 2005).



Additionally, biases in precipitation estimates, scaling issues, and limitations in model structure can further complicate the relationship between precipitation improvements and river discharge accuracy (Butts et al., 2004).

It is worth noting that using a distributed hydrological model could better capture the benefits of high-resolution climate model outputs, as lumped models may not reflect the spatial heterogeneity present in such data. However, for the sake of

simplicity and to enhance the generalizability of the results, a lumped model was used in this study. That said, a recommended next step would be to repeat the analysis using a distributed model, such as HydroTel (Fortin et al., 2001). Both climate model outputs (CRCM6/GEM5-2.5km and CRCM6/GEM5-12km) tend to an overestimation of peak flows, though CRCM6/GEM5-2.5km generally exhibits lower bias. These findings emphasize the complex relationship between climate model resolution and hydrological performance.

The differences in precipitation representation have direct implications for hydrological modeling. The KGE values (Fig. 5) indicate that both climate models produce lower river discharge accuracy than the reanalysis dataset. This is expected, as climate models are not designed to reproduce historical events but rather to represent general climatic characteristics (mean, variability and extremes). Differences in model forcings, parameterizations, and temporal mismatches between observed and simulated datasets may contribute to these discrepancies. A lower metric for hydrological modeling performance simply

indicates this.

Peak flow analyses (Fig. 7) further highlight the influence of climate model resolution on hydrological extremes, particularly in terms of the accuracy of peak flow estimations. For Boyer Sud and Bras d'Henri, the two basins with the strongest normalized peak flows (refer to Fig. 6), all simulations underestimated the most intense observed peak flows. This is likely due to a combination of factors, including errors in precipitation estimation, limitations of the hydrological model, and

potential biases in observational records. However, the finer spatial resolution of CRCM6/GEM5-2.5km provides a more accurate representation of peak flows compared to CRCM6/GEM5-12km, reinforcing the added value of high-resolution climate modeling for flood risk assessment.

The influence of basin size on simulated river discharge accuracy is evident for both the KGE (Fig. 5) and peak flow analyses (Fig. 6-9). Larger basins exhibit more stable hydrological responses, with simulated river discharge values

converging more closely with observations. This is expected, as larger basins integrate precipitation inputs over broader areas, reducing the impact of localized errors. In contrast, smaller basins are more sensitive to high precipitation intensities, short-duration rainfall events, making accurate precipitation representation crucial for reliable flood simulation. The basin's location also affects the biases. As an example of the combined effects of basin size and location, the results for certain basins, for example, those from the 'near Montreal city' group, show strong similarities across different simulations. This is

evident from the overlapping interquartile ranges and the proximity of median bias values in the boxplots (Fig. 7). This consistency may be attributed to similar precipitation forcing, basin characteristics, or hydrological model performance in these regions. Similar patterns have been observed in other basin groups as well (groups' detail in Sect. 2.1).

An interesting and counterintuitive result emerges when comparing the peak flows simulated using the CRCM6/GEM5-2.5km and CRCM6/GEM5-12km climate model outputs. Although extreme precipitation events are generally more intense



in the CRCM6/GEM5-2.5km simulations across the study area (see Fig. 4), certain basins, such as Des Hurons and
       L'Acadie, exhibit higher peak flows when forced with the CRCM6/GEM5-12km data (see Fig. 7). This apparent
       contradiction can be explained by the difference in spatial resolution between the two models. The grid cells in the 12 km
       version are over 20 times larger in area than those in the 2.5 km version. Given that extreme summer-fall precipitation events
       often occur on very localized scales (less than a few kilometers), the higher-resolution model tends to confine extreme
rainfall to a limited number of small grid cells. In contrast, the coarser 12 km model tends to spatially average or spread
       these events over a much larger area, effectively simulating a broader extent of high rainfall. As a result, the total volume of
       water contributing to runoff—and thus the simulated peak flow—can be greater in the 12 km model, despite the finer-
       resolution model capturing more intense localized precipitation.

## 5.3 Implications for climate adaptation and flood risk management

The findings of this study have significant implications for climate adaptation and risk management. The underestimation of
       extreme precipitation and peak flows by coarse-resolution models suggests that relying solely on traditional climate datasets
       may lead to under preparedness in flood-prone regions. High-resolution climate simulations, such as CRCM6/GEM5-2.5km
       could improve flood risk assessments and design resilient infrastructure. Moreover, the differences between summer and fall
       precipitation highlight the seasonal dependence of climate model performance. This suggests that climate impact studies
should consider seasonal variability when assessing model reliability for different applications. Overall, this study reinforces
       the need for high-resolution climate models in hydrological applications and highlights the importance of continued
       advancements in regional climate modeling to improve the simulation of extreme weather events.

       6.Conclusion

       In this study we focused on warm-season extreme precipitation events (summer-fall) in southern Quebec and assessed the
performance of two resolution outputs from the most recent Canadian regional climate model, CRCM6/GEM5, at 12 km and
       2.5 km resolutions. CRCM6/GEM5-2.5km showed significantly better agreement with weather station data in terms of both
       the frequency of rainfall events and the intensity of extreme precipitation at an hourly timescale. This marks a notable
       improvement over CRCM6/GEM5-12km and suggests that higher-resolution climate models can provide more reliable
       information for assessing extreme rainfall and floods in the region.

Beyond precipitation analysis, our study explored whether the added value of CRCM6/GEM5-2.5km extends to hydrological
       modeling, specifically in simulating peak hourly flows. Using a hydrological model at an hourly time step, we assessed the
       ability of both climate model outputs to reproduce observed peak flows during the period 2001-2018 for 11 basins across
       southern Quebec. The results shown that for most basins, simulated peak flows using CRCM6/GEM5-2.5km data exhibited
       lower biases and greater accuracy compared to those simulated with CRCM6/GEM5-12km. This suggests that higher-
resolution climate data can improve not only precipitation estimates, but also hydrological simulations, which are crucial for
       flood risk assessment and water resource management.





These findings have important implications for climate adaptation strategies, particularly in regions prone to extreme rainfall and flooding. By improving our ability to simulate and project extreme events at finer spatial and temporal scales, high-resolution climate models can provide valuable input for decision-makers in government, urban planning, and the insurance industry. In the context of Quebec, where climate change is expected to bring more frequent and intense storms, adopting such advanced modeling techniques can enhance preparedness and resilience against future extreme events. This study is a first step toward aligned with incorporating new climate model outputs (CRCM6/GEM5-2.5km) for being used in the production of Atlas hydroclimatique du Quebec. The Atlas hydroclimatique du Québec is an interactive mapping tool developed by the Direction principale de l'expertise hydrique (DPEH). It provides a comprehensive overview of the current and projected future hydrological regimes of rivers in southern Quebec, with a focus on the impacts of climate change.

Overall, this study highlights the added value of using finer climate model resolutions for extreme event analysis and hydrological modeling. As climate risks continue to escalate, leveraging high-resolution climate data will be essential for improving disaster preparedness, infrastructure resilience, and policy planning in the face of a rapidly changing climate.

As a next step, it is necessary to assess the differences between the two climate model resolutions in terms of how each projects future precipitation scenarios. This comparison will provide critical insights into the influence of model resolution on projected hydroclimatic extremes, which is essential for accurately estimating associated flood risks. Understanding these differences will help to identify potential biases or uncertainties in flood projections and to improve the reliability of climate-informed water resource and risk management strategies.

**Acknowledgements**

This study was funded by the Réseau Inondations InterSectoriel du Québec (RIISQ) and Ouranos.

The authors would like to thank the anonymous reviewers for their insightful and constructive comments that helped improve the quality of this manuscript.

We also extend our gratitude to Katja Wagner and the ESCER Centre for their technical assistance in accessing climate model data (ERA5-Land, RDRSV2.1) and for preparing Figures 1a and 1b.

RDRSv2.1 data (which is also called Canadian Surface Reanalysis (CaSR)) are available through the CaSPAr webpage at the following address: https://github.com/julemai/CaSPAr/wiki/How-to-get-started-and-download-your-first-data

River discharge data at hydrometric stations were provided by the Ministère de l'Environnement, de la Lutte contre les changements climatiques, de la Faune et des Parcs (MELCCFP), and are available at:

https://www.cehq.gouv.qc.ca/hydrometrie/historique_donnees/fiche_instantanee.asp?NoStation=XXXXXX,

where "XXXXXX" corresponds to the station numbers listed in Table 1.

Weather station data were obtained from Environment and Climate Change Canada (ECCC) and are available at:

https://climate.weather.gc.ca/historical_data/search_historic_data_e.html.



We acknowledge the use of the ERA5-Land reanalysis dataset produced by Muñoz-Sabater et al. (2021), downloaded by the
ESCER Centre staff from the Copernicus Climate Change Service (C3S) Climate Data Store:
https://cds.climate.copernicus.eu/datasets/reanalysis-era5-land.

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
