# Peer review of "Can high-resolution convection-permitting climate models improve flood simulation in southern Quebec watersheds?"

_EGUsphere, 2025_

## Author Comment (AC1)

**The authors evaluate the perfomance of a 12-km RCM and a 2.5-km CPRCM with respect to reproduction of primarily observed precipitation (rainfall) and discharge extremes in Quebec. Is is found (claimed) that the CPRCM better reproduces both types of extremes. While the topic is relevant and interesting, the methods appear overall well selected and applied, the presentation is neat, I cannot recommend publication of the manuscript in its present form. In the following I will explain why.**

**General comments**

- **The paper tries to argue that the CPRCM does a much better job than the RCM, but this is, in my view, not (well) supported by the results. Sometimes it is clear that the CPRCM greatly overestimates the extreme rainfall (Fig. 3a, Fig. 4a), but this is basically neglected. Sometimes argumentation is based on visual inspection (Figs. 6-9) or a few numbers (Table 3) but without testing whether differences are statistically significant (which is really important, especially as you sometimes work with very small data sets). OK, it may be that the CPRCM does a (significantly) better job than the RCM, but the current results do not prove that.**

We thank the reviewer for this comment. Our response has two parts: Part 1 addresses the need for additional quantitative assessment, and Part 2 provides explanations regarding the apparent overestimation of precipitation by the CPRCM relative to station observations.

Part 1:

We agree that additional quantitative assessments and statistical tests are necessary to better demonstrate the significance of the differences between CPRCM and other RCM simulations. To address this point, we conducted additional analyses focusing on extreme precipitation (99.99th percentile) at station locations. First, we compared model outputs with station observations using standard performance metrics, RMSE, mean bias, and correlation, to quantify how well each model reproduces extreme precipitation during the summer and fall seasons. Station-based estimates were compared with the corresponding values extracted at the same grid points from the model simulations (CRCM6/GEM5 at 12 km and 2.5 km resolution) and from the reanalysis dataset (RDRSv2.1). The RMSE was computed using values from all stations, while the mean bias was calculated as the

average of station-wise biases for each dataset. Correlation coefficients were also computed using values from all stations.

Overall, CRCM6/GEM5 at 2.5 km resolution shows the best performance across all metrics (see Tables below). RMSE values are lowest for CRCM6/GEM5–2.5 km, even lower than those obtained from the reanalysis dataset. In addition, CRCM6/GEM5–2.5 km exhibits a smaller mean bias relative to station-based extreme precipitation estimates. Furthermore, results show a higher correlation with station data during summer. In fall, all datasets exhibit low correlation.

| | RMSE (mm) | |
| --- | --- | --- |
| | Summer | Fall |
| CRCM6/GEM5-2.5KM | **5.63** | **14.28** |
| CRCM6/GEM5-12KM | 13.42 | 15.72 |
| RDRSv2.1 | 8.74 | 14.87 |

| | averaged bias | |
| --- | --- | --- |
| | Summer | Fall |
| CRCM6/GEM5-2.5KM | **0.06** | **0.11** |
| CRCM6/GEM5-12KM | -0.47 | -0.27 |
| RDRSv2.1 | -0.24 | -0.16 |

| | Correlation | |
| --- | --- | --- |
| | Summer | Fall |
| CRCM6/GEM5-2.5KM | **0.37** | -0.01 |
| CRCM6/GEM5-12KM | 0.32 | 0.01 |
| RDRSv2.1 | 0.25 | 0.06 |

Considering previous assessments, we formally evaluate whether the differences between the two model configurations are statistically significant by applying the

Wilcoxon signed-rank test to the station-based 99.99th percentile of hourly precipitation, comparing the two simulations. These additional analyses allow for a more rigorous assessment of the added value of the CPRCM relative to the coarser-resolution RCM and strengthen the interpretation of the results. The table below presents the results of the Wilcoxon signed-rank test (significance level of 5%, p-value = 0.05). The null hypothesis states that the median of the differences between the 99.99th percentile hourly precipitation simulated by CRCM6/GEM5 at 2.5 km and 12 km resolution is zero, implying no systematic difference between the two resolutions. The test results (H = 1) indicate rejection of the null hypothesis. Specifically, the null hypothesis is rejected for both summer and fall, demonstrating that the differences between the two simulations are statistically significant in both seasons.

|         | p-value | H |
|---------|---------|---|
| Summer  | 0.00    | 1 |
| Fall    | 0.00    | 1 |

Part 2:

Nevertheless, we do not fully agree that CPRCM greatly overestimates summer values, as suggested by Figs. 3a and 4a. In Fig. 3a, the CPRCM histogram matches the station data reasonably well. It is true that the vertical line representing the 99.99th percentile is substantially higher than the observed one; however, this likely results from estimating an empirical 99.99th percentile, which is highly sensitive to a small number of extreme values in the tail of the distribution. In addition, the station shown in the paper is only one example. When considering other stations (with histograms provided in the supplementary material), the difference between the station-based and CPRCM-based 99.99th percentiles is not consistently large. Furthermore, the RMSE and mean bias of the 99.99th percentile indicate that CPRCM has a lower RMSE (i.e., better performance) than the other datasets, as well as a smaller bias. Regarding Fig. 4a, some overestimation is also present, but its magnitude remains limited.

- **Some very important aspects are not taken into account, notably the impact of spatial resolution and the impact of climate model bias, both having a huge impact on both rainfall and discharge (extremes). They are briefly mentioned but without being investigated, which makes the significance of the results virtually impossible to judge, in my opinion.**

We thank the reviewer for this important comment. We fully agree that both spatial resolution and climate model bias can strongly affect simulated precipitation and discharge, particularly for extremes.

The impact of spatial resolution is investigated by comparing simulations performed with CRCM6 at 2.5 km and 12 km, driven by the same large-scale forcing and using consistent physical parameterizations. Differences between the two simulations therefore primarily reflect the effect of spatial resolution and the representation of convective processes.

To make this point clearer, we will revise Section 5 to explicitly discuss the role of spatial resolution and its expected influence on extreme precipitation intensities and spatial coherence.

We agree that climate model biases play a crucial role in the simulation of precipitation and discharge extremes. Numerous studies have shown that increasing spatial resolution, particularly toward convection-permitting scales, tends to increase the intensity of short-duration precipitation extremes due to an improved representation of convective processes and reduced reliance on convective parameterization (e.g., Prein et al., 2015; Kendon et al., 2017; Ban et al., 2021). This increase in extreme precipitation intensity is often associated with a reduction of the well-documented underestimation of sub-daily extremes in coarser-resolution regional climate models, although biases are not fully eliminated. In this study, we therefore interpret differences between the 2.5 km and 12 km simulations primarily as resolution-related effects, while acknowledging that residual model biases remain.

The second paragraph (line 546) of section 5.1 will be replaced by following paragraph:

'*The comparison of precipitation intensity distributions and spatial patterns (Figs. 3 and 4) highlights a key limitation of coarser-resolution climate models, namely their tendency to underestimate short-duration, high-intensity rainfall due to an inadequate representation of fine-scale convective processes. In contrast, CRCM6/GEM5-2.5 km better captures both the intensity and localized nature of*

*extreme precipitation, particularly in summer when convective storms dominate, consistent with previous studies showing that finer spatial resolution improves the simulation of convective systems (e.g., Ban et al., 2014; Prein et al., 2015, 2020; Kendon et al., 2014; Ban et al., 2021). The larger extremes simulated at 2.5 km should not be interpreted as the absence of model bias; rather, convection-permitting models are known to produce higher short-duration precipitation intensities partly due to reduced spatial smoothing and improved process representation, which often alleviates, but does not eliminate, the underestimation of sub-daily extremes in coarser models. Because both simulations use the same large-scale forcing and model framework, the observed differences are mainly due to resolution effects rather than independent model biases. However, residual biases in the absolute magnitude of extremes may still affect hydrological impact assessments. Consequently, while the 2.5 km configuration demonstrates clear added value for applications such as flood risk analysis, future work using bias-corrected simulations or multi-model ensembles would help further disentangle the respective roles of spatial resolution and model bias.'*

- **The text is overall well written, but with far too much of superfluous information (things that one must assume is known to HESS readers) and repetition. To me it has the feel of a (good) student essay, but not on the level of a scientific paper in HESS. Some examples:**

  - **The description of GCMs and RCMs in the introduction (40-55) is known to readers.**

  - **Readers know how KGE works (248-250), what a CDF is (296-299), how box plots should be interpreted (447-451), and other similar examples.**

Thank you for these fair points. The text will be revised to reduce repetition by shortening some explanations and making the discussion more concise and straightforward.

  - **Section 5 (Discussion) is in my view mainly a rather long summary of the study, mainly consisting of repetitions and with few real conclusions, other than general statements.**

We thank the reviewer for this suggestion. We agree that the original version of Section 5 contains a substantial summary of the paper. To improve clarity and focus, lines 568-571 will be replaced by following paragraph:

'*The use of a lumped hydrological model in this study was a deliberate and conservative choice aimed at maintaining the methodology simple and enhancing the reproducibility. By design, lumped models do not explicitly take advantage of the spatial organization of precipitation, and therefore the fine-scale rainfall structures simulated by convection-permitting climate models (CPRCMs) are not directly translated into the runoff generation process. Consequently, the hydrological response primarily reflects differences in precipitation intensity and temporal variability, while spatially localized convective extremes are averaged at the catchment scale when used in a lumped model. This aggregation may attenuate or even mask potential CPRCM added value, and likely contributes to cases where the 12 km simulation yields higher KGE values than the 2.5 km simulation. Moreover, the fact that hydrological improvements are nevertheless observed using CPRCM outputs within a lumped modeling framework indicates that these results represent a lower-bound estimate of the added value of high-resolution climate simulations. This suggests that greater gains in the simulation of extreme flows could be achieved in future studies through the use of distributed hydrological models, such as Hydrotel (Fortin et al., 2001), which are better suited to exploit spatial rainfall heterogeneity. Overall, these findings underscore that hydrological added value depends not only on atmospheric model resolution, but also on the consistency between climate and hydrological model structures, and that the limitations of the lumped approach should be explicitly considered when interpreting improvements in flood simulations*.'

**Specific comments:**

- **164: Is this supposed to be a separate section?**

Thanks for your attention, yes, it will be corrected.

- **270: 18 annual maxima is not much to work with, consider peak-over-threshold instead.**

.This is a valid point. To increase the sample size for the analysis, a peak-over-threshold (POT) approach will be adopted. In the POT method, a minimum separation of three days between consecutive peak events will be imposed to ensure the independence of the extracted events.

- **315: Should be (Fig. 3a, 3c, 3f), I think.**

This is correctly mentioned, since it was supposed to refer to the summer season plots for both CRCM6/GEM5 configurations.

- **358-363: There is no need to repeat the figure caption in text, same happens also elsewhere. And how to visually interpret the match bewteen histograms is trivial. In addition, you do not need to explain the colour legends in the captions; we understand from the panels.**

This is a fair point, it will be corrected in revised version

- **454: One "generally" too much here.**

Thanks, the first "generally" will be removed

---

## Author Comment (AC2)

**The manuscript entitled "Can high-resolution convection-permitting climate models improve flood simulation in southern Quebec watersheds?" investigates whether CPRCM simulations provide added value for flood simulation in southern Quebec, with focus on extreme summer and fall rainfall events. The topic is relevant and the objectives of the study are clear. The manuscript is also well organized and written. The comparison between the 12 km and 2.5 km CRCM6/GEM5 simulations, and try to link precipitation extremes to hydrological responses, are interesting to both the regional climate and hydrological communities. However, several important issues need to be addressed before the conclusions can be fully supported.**

**1.  The study aims to assess the added value of CPRCM simulations. In this context, the choice of a lumped hydrological model raises concerns. The hydrological model used in this study (GR5dt) is a conceptual and lumped model with only elevation bands. Given that the main added value of CPRCMs lies in their improved representation of the spatial patterns of intense precipitation, the use of a lumped model may substantially limit the ability to assess this added value. For example, Figure 5 shows that for many basins the KGE driven by RCM-12 km is higher than that driven by CPRCM-2.5 km, which may partly reflect this modeling choice. While this limitation is mentioned in the discussion, the authors should better justify the choice of model and more clearly acknowledge the associated limitations when interpreting the improvements in flood simulations.**

We thank the reviewer for this important comment. We agree that the use of a conceptual lumped hydrological model such as GR5dt limits the ability to fully exploit the spatial information provided by CPRCM simulations.

The choice of GR5dt was motivated by the objectives of this study, which focus on assessing the sensitivity of simulated flood responses to differences in precipitation intensity and temporal structure between CPRCM (2.5 km) and coarser-resolution RCM (12 km) outputs, rather than on explicitly resolving spatial runoff generation processes within catchments. Importantly, only a limited number of hydrological models are suitable for simulations at the hourly time scale, and GR5dt is one of the few lumped models specifically designed and validated for this purpose. GR models have been widely used and shown to perform robustly for flood simulations at daily and sub-daily time scales across a range of catchment sizes, while requiring limited calibration parameters, which was essential for ensuring a consistent comparison across multiple basins.

We fully acknowledge that lumped models cannot explicitly represent the spatial variability of rainfall and runoff processes within a basin. However, the catchments considered in this study are generally small and characterized by relatively flat topography, conditions under which lumped modeling approaches are commonly considered appropriate.

Nevertheless, part of the added value of CPRCMs, particularly the improved representation of fine-scale spatial precipitation patterns, may not be fully translated into improved simulated river discharge when using a lumped framework. This limitation may partly explain why, for several basins, the KGE driven by the 12 km RCM exceeds that obtained using the 2.5 km CPRCM (Fig. 5).

To address this point more explicitly, lines 568-571 will be replaced by following paragraph:

*'The use of a lumped hydrological model in this study was a deliberate and conservative choice aimed at maintaining the methodology simple and enhancing the reproducibility. By design, lumped models do not explicitly take advantage of the spatial organization of precipitation, and therefore the fine-scale rainfall structures simulated by convection-permitting climate models (CPRCMs) are not directly translated into the runoff generation process. Consequently, the hydrological response primarily reflects differences in precipitation intensity and temporal variability, while spatially localized convective extremes are averaged at the catchment scale when used in a lumped model. This aggregation may attenuate or even mask potential CPRCM added value, and likely contributes to cases where the 12 km simulation yields higher KGE values than the 2.5 km simulation. Moreover, the fact that hydrological improvements are nevertheless observed using CPRCM outputs within a lumped modeling framework indicates that these results represent a lower-bound estimate of the added value of high-resolution climate simulations. This suggests that greater gains in the simulation of extreme flows could be achieved in future studies through the use of distributed hydrological models, such as Hydrotel (Fortin et al., 2001), which are better suited to exploit spatial rainfall heterogeneity. Overall, these findings underscore that hydrological added value depends not only on atmospheric model resolution, but also on the consistency between climate and hydrological model structures, and that the limitations of the lumped approach should be explicitly considered when interpreting improvements in flood simulations.'*

In addition, the manuscript states that "KGE values slightly increase as basin size

increases", whereas the figure shows substantial variability from left to right. Other factors may influence the results, such as elevation-related precipitation biases. An elevation-dependent analysis (e.g., bias or performance metrics versus basin mean elevation) could be very informative, especially given known elevation-dependent biases in precipitation.

We thank the reviewer for this observation. We agree that the statement suggesting that KGE values increase with basin size was too strong given the substantial variability observed across basins. Our intention was to indicate a weak tendency rather than a systematic relationship. To avoid overinterpretation, we will revise the text to clarify that basin size alone does not explain the variability in model performance.

We further agree that additional factors, such as elevation-related precipitation biases, may influence the hydrological results. Orographic effects and elevation-dependent biases are known to affect both precipitation magnitude and phase, and their impact may differ between the 2.5 km and 12 km simulations. In the current study, these effects are not explicitly analyzed, and we therefore interpret the hydrological results as reflecting an interplay between basin characteristics, precipitation forcing, and model structure rather than a simple dependence on basin size.

We will revise the Discussion to acknowledge the potential role of elevation and to highlight that an elevation-dependent evaluation (e.g., relating bias or performance metrics to basin mean elevation) would provide additional insight into the hydrological added value of CPRCM simulations. Such analyses are identified as an important direction for future work.

**2.  Table 3 presents useful results. However, in Figure 7, several basins do not show a clear added value in the boxplots of peak flow bias (e.g., small basins such as Bras d'Henri, as well as larger basins such as Eaton, Au Saumon, and Etchemin). Additional explanation would be helpful here. In addition, it would be useful to clarify whether the peak flow bias reported in Table 3 is defined consistently with that shown in Figure 9.**

We thank the reviewer for this careful reading of the results. We agree that the added value of the CPRCM-driven simulations is not systematic across all basins, as illustrated by the boxplots of peak flow bias in Figure 7. This heterogeneity reflects differences in basin characteristics, scale, and the interaction between precipitation forcing and the hydrological modeling framework.

For small basins such as Bras d'Henri, flood response is highly sensitive to localized precipitation extremes and their spatial positioning within the catchment. While the CPRCM provides a more realistic representation of localized convective precipitation, this added spatial detail is largely averaged out when used as input to the lumped hydrological model, which may limit or even counteract potential improvements in peak flow simulation.

For larger basins (e.g., Eaton, Au Saumon, and Etchemin), the basin-averaging effect becomes even stronger, and flood peaks are influenced by a combination of spatial aggregation, routing processes, and also on how runoff from different parts of the basin reaches the outlet at similar or different times i.e. temporal synchronization of runoff contributions. In such cases, smoother precipitation fields from the coarser-resolution RCM may sometimes lead to comparable or better peak flow bias metrics, despite a less realistic representation of precipitation extremes at smaller scales.

Yes, the peak flow bias reported in Table 3 is calculated using the same peak flow values as those shown in Figure 9. To avoid any ambiguity, we will clarify this definition in the manuscript.

**3.   Biases in simulated precipitation strongly affect hydrological model performance. I understand that no bias correction was applied to the climate model forcings before driving the hydrological model. However, the potential impact of precipitation biases on the flood simulation results, should be discussed more explicitly.**

We thank the reviewer for this important comment. We agree that biases in simulated precipitation can strongly influence hydrological model performance, particularly for flood simulations. In this study, no explicit bias correction was applied to the climate model forcings, as the objective was to assess the relative hydrological response to precipitation generated by climate models at different spatial resolutions within a consistent modeling framework. Bias correction would add another layer of complexity that might make the results difficult to assess. Furthermore, bias correcting subdaily precipitation data at a 2.5 km resolution is not a trivial matter, especially with regards to correcting extremes as well as the rarity and uncertainty related to the reference dataset that would need to be used.

We acknowledge that biases in precipitation magnitude, intensity, and temporal structure may directly affect simulated flood peaks and performance metrics. In particular, biases in short-duration extreme precipitation may lead to over- or

underestimation of peak flows, depending on basin characteristics and hydrological model sensitivity. While both the 2.5 km and 12 km simulations are affected by model biases, their impacts may differ across basins and seasons, contributing to the heterogeneous hydrological performance observed in the results.

To address this point more explicitly, we will expand the Discussion to clarify how precipitation biases may influence the flood simulation results and to emphasize that the hydrological outcomes should be interpreted primarily in a comparative sense rather than as bias-free representations of observed floods. The application of bias-correction techniques and their interaction with model resolution are identified as important directions for future work.

**4.   The manuscript suggests that CPRCM outputs improve flood simulations and can be useful for risk management strategies. It remains unclear whether the authors imply that CPRCM outputs can be used directly for hydrological applications. If so, under what conditions (e.g., bias correction, calibration)? A clearer discussion of the applicability and limitations of using CPRCM outputs for hydrological modeling would strengthen the paper.**

We thank the reviewer for this important comment. We clarify that we do not argue that CPRCM outputs are universally bias-free; however, our results indicate that CPRCM simulations show closer agreement with station observations than the coarser-resolution RCM, particularly for extreme precipitation. In this context, we believe that CPRCM outputs can be used directly for flood simulations to provide a more realistic estimation of flood magnitudes, especially in comparative or exploratory studies. Bias adjustment of the bulk of the distribution is generally more straightforward than that of the extremes. If, for example, the biases in precipitation extremes simulated by the CPRCM are shown to be acceptable, it may be justified to use the data within a hybrid correction framework in which the bulk of the distribution is bias-corrected (to preserve the annual water balance), while the extremes are left unadjusted. This highlights the importance of improving our understanding of biases in extreme precipitation. However, the development of such a correction approach is beyond the scope of the present study.

At the same time, we acknowledge that systematic precipitation biases may still be present and that bias correction can further improve hydrological performance, particularly for applications requiring accurate absolute flood quantification or in cases where the biases are too large to ignore.

Bias correction is therefore recommended for operational or risk management applications, but its application is not strictly required to assess relative flood behavior or to benefit from the improved representation of precipitation extremes provided by CPRCMs.

We will revise the Discussion to clarify the conditions under which CPRCM outputs may be used directly and to more clearly distinguish between exploratory flood analyses and applications requiring high accuracy, such as flood risk management.

**Overall, the study addresses an important question, and provides useful information for hydroclimate research.**